# Phylogenetic Revision and Patterns of Host Specificity in the Fungal Subphylum Entomophthoromycotina

**DOI:** 10.3390/microorganisms10020256

**Published:** 2022-01-24

**Authors:** Lars Möckel, Karen Meusemann, Bernhard Misof, Volker U. Schwartze, Henrik H. De Fine Licht, Kerstin Voigt, Benjamin Stielow, Sybren de Hoog, Rolf G. Beutel, Jan Buellesbach

**Affiliations:** 1Jena Microbial Resource Collection, Leibniz Institute for Natural Product Research and Infection Biology, 07745 Jena, Germany; lars.moeckel@schule.thueringen.de (L.M.); volker.schwartze@uni-jena.de (V.U.S.); kerstin.voigt@hki-jena.de (K.V.); 2Institute of Microbiology, Faculty of Biological Sciences, Friedrich Schiller University Jena, 07743 Jena, Germany; 3Institute of Zoology and Evolutionary Research, Friedrich Schiller University Jena, 07743 Jena, Germany; rolf.beutel@uni-jena.de; 4Zoological Research Museum Alexander Koenig, Leibniz Institute for the Analysis of Biodiversity Change, 53113 Bonn, Germany; k.meusemann@leibniz-zfmk.de (K.M.); b.Misof@leibniz-zfmk.de (B.M.); 5Department of Plant and Environmental Science, University of Copenhagen, 1871 Frederiksberg, Denmark; hhdefinelicht@plen.ku.dk; 6Center of Expertise in Mycology, Radboud University Medical Center, 6525 GA Nijmegen, The Netherlands; benjaminstielow@gmail.com (B.S.); Sybren.deHoog@radboudumc.nl (S.d.H.); 7Westerdijk Fungal Biodiversity Institute—KNAW, 3584 CT Utrecht, The Netherlands; 8Institute for Evolution & Biodiversity, University of Münster, 48149 Münster, Germany

**Keywords:** terrestrial fungi, Zygomycota, Zygomycetes, zygosporic fungi, molecular phylogeny, species diversity, evolutionary interactions

## Abstract

The Entomophthoromycotina, a subphylum close to the root of terrestrial fungi with a bias toward insects as their primary hosts, has been notoriously difficult to categorize taxonomically for decades. Here, we reassess the phylogeny of this group based on conserved genes encoding ribosomal RNA and RNA polymerase II subunits, confirming their general monophyly, but challenging previously assumed taxonomic relationships within and between particular clades. Furthermore, for the prominent, partially human-pathogenic taxon *Conidiobolus*, a new type species *C. coronatus* is proposed in order to compensate for the unclear, presumably lost previous type species *C. utriculosus* Brefeld 1884. We also performed an exhaustive survey of the broad host spectrum of the Entomophthoromycotina, which is not restricted to insects alone, and investigated potential patterns of co-evolution across their megadiverse host range. Our results suggest multiple independent origins of parasitism within this subphylum and no apparent co-evolutionary events with any particular host lineage. However, Pterygota (i.e., winged insects) clearly constitute the most dominantly parasitized superordinate host group. This appears to be in accordance with an increased dispersal capacity mediated by the radiation of the Pterygota during insect evolution, which has likely greatly facilitated the spread, infection opportunities, and evolutionary divergence of the Entomophthoromycotina as well.

## 1. Introduction

The subphylum Entomophthoromycotina originated from one of the largest and oldest known radiation events of terrestrial fungi [1]. A Devonian origin has been suggested for this taxon around 405 ± 90 mya [2], which implies that they existed in terrestrial environments together with early winged insects [3]. The Greek name is a composite of entomon (insect), phthor (destroyer), and mycota (fungi). It refers to the best-known and most diverse host taxon of these organisms, insects, and to the widely described entomopathogenic potential in this fungal lineage (Figure 1).

Interestingly, although most of the research focused on Entomophthoromycotina is dedicated to obligate or facultative parasites of insects [4], the majority of extant species in this subphylum are soil living and saprotrophic [5,6]. It is currently unclear how the obligate and often host-specific entomopathogenic lifestyle of some of the entomophthoromycotinan taxa evolved [7]. Some of the molecular mechanisms associated with adaptations for an entomopathogenic lifestyle have recently been unraveled [8,9,10], and molecular phylogenetic analyses seem to corroborate multiple independent origins of entomopathogenicity within this subphylum [11]. In contrast to obligate insect-pathogenic species, some primarily saprotrophic taxa apparently possess the capacity to switch to parasitism, also causing infections in humans [12,13]. For instance, species of the genera *Basidiobolus* and *Conidiobolus* are well-known human parasites [14]. However, it is unclear how frequently these switches from a saprotrophic to a parasitic lifestyle occurred.

Presently, about 70 of the 321 known Entomophthoromycotina species have been documented to be capable of infecting a broad range of host taxa, from arthropods to humans [11,14]. The host range varies distinctly between generalist genera such as *Basidiobolus* and *Conidiobolus* on the one hand and obligatory entomopathogenic specialists on the other. Some species are capable of infecting insect hosts across several orders, whereas others naturally only infect individuals of a specific family, genus, or even species. Apparently, host-specific pathogenic fungi do not switch hosts at all and also lack the potential to cause opportunistic infections [14,15,16]. This renders host-specific entomopathogenic members of this subphylum particularly suited as biological control agents in integrated pest management programs. In contrast to chemical control (i.e., pesticides), biological pest control does not only have a higher potential to keep pest species from ever reaching an economically damaging level but also bears considerable advantages for human health, agricultural quality, and environmental sustainability [17,18,19]. However, to properly assess taxa within the Entomophthoromycotina with targeted host specificity for particular pest species, a clearer phylogenetic understanding of this clade will be indispensable.

The difficulties in separating genera within the Entomophthoromycotina are mirrored by the shifting position of this taxon within the Zygomycota, a group notoriously difficult to assess taxonomically and phylogenetically, which had previously been classified as a superordinated phylum [20,21]. However, recent molecular and morphological studies clearly rejected the previously assumed monophyly of the Zygomycota [1,22].

Similarly, Entomophthoromycotina is a taxon with an uncertain phylogenetic basis [23], previously assumed to be represented by at least five unrelated clades [24]. At the genus level, the aforementioned partially human-pathogenic taxon *Conidiobolus* constitutes a prime example of the difficulty in taxonomically characterizing entomophthoromycotinan lineages. First, being grouped into four clades that appear to be paraphyletic to each other [25], it has consequently been suggested to reject *Conidiobolus* and its three subgenera, *Conidiobolus*, *Capillidium*, and *Delacroixia* [26]. The latter two subgenera were then elevated to genus level with the additional description of two new genera, *Microconidiobolus* and *Neoconidiobolus* [25]. Clearly, the taxonomic assessment of this prominent taxon which has gained notoriety for its capability to switch from a primarily saprobic to a parasitic lifestyle requires a thorough revision.

We also have very limited general knowledge of the evolutionary history of the Entomophthoromycotina, their diversity, as well as how and when switches of lifestyles occurred. The number of extant insect-pathogenic as well as saprobic lineages differs widely between the established clades within this subphylum [27]. To gain a more holistic view on the evolutionary history, diversity as well as host range and specificity of the Entomophthoromycotina, we conducted a large-scale phylogenetic analysis of 159 entomophthoromycotinan strains based on sequence data of three genes [28], two that encode large and small subunits of the ribosome (LSU 28S and SSU 18S, respectively), and one that encodes an RNA polymerase II subunit (RPB2). We combine our phylogenetic analysis with available information on the host range of the investigated entomophthoromycotinan taxa. Specifically, we ask (1) whether the postulated monophyly of different Entomophthoromycotina subclades can be confirmed and (2) whether patterns of diversity and co-evolution between the pathogenic Entomophthoromycotina taxa and their hosts, particularly insects, are detectable. Therefore, we mainly focus on the degree of radiation in insect-pathogenic lineages related to the extent of host specialization.

## 2. Materials and Methods

### 2.1. Data Acquisition of Fungal Strains

For phylogenetic analysis, we used sequence data retrieved from the National Center for Biotechnology Information (NCBI) Bethesda (MD, USA) and from the Westerdijk Fungal Biodiversity Institute (formerly: Centraalbureau voor Schimmelcultures (CBS)—KNAW Fungal Biodiversity Center) Utrecht (NL). Data acquired from the Westerdijk Fungal Biodiversity Institute was obtained during a long-term sequencing project targeting the entire fungal collection. The corresponding molecular methods are detailed in [29,30].

We collected nucleic acid sequences encoding the ribosomal RNA (LSU: 69 species in 147 strains; SSU: 54 species in 91 strains) and the DNA-directed RNA polymerase II subunit (RPB2) from a total of 71 strains from 36 species of the Entomophthoromycotina. Sequences from the following sources were included: AFTOL (Assembling The Fungal Tree of Life, NSF, USA), ARSEF (Agricultural Research Service, Collection of Entomopathogenic Fungal Cultures, Ithaca, NY, USA), ATCC (American Type Culture Collection, Manassas, VA, USA), CBS (Centraalbureau voor Schimmelcultures, Utrecht, NL, now: Westerdijk Fungal Biodiversity Institute Utrecht, NL), FSU (Friedrich-Schiller-Universität Jena, Jena Microbial Resource Collection, GER), NRRL (Northern Regional Research Laboratories, now: National Center for Agricultural Utilization Research, Agricultural Research Service, Culture Collection, Ithaca, NY, USA) and RCEF (Research Center for Agricultural Fungi, Hefei, China). The following representatives of the Mucorales, a sister group within the zygosporic fungi, were used as outgroup taxa: *Rhizopus oryzae*, *Rhizopus azygosporus*, *Rhizomucor pusillus*, *Rhizomucor miehei*, *Rhizomucor variabilis,* and *Parasitella parasitica* https://www.ncbi.nlm.nih.gov/nuccore/ (accessed on 15 April 2013). Additional information on all strains used in the present study are indicated in Appendix A.

The following genera were not included in the analysis due to incomplete or missing sequence data: *Meristacrum* (Meristacraceae), *Tabanomyces* (Meristacraceae), *Apterivorax* (Neozygitaceae), *Thaxterosporium* (Neozygitaceae), *Ancylistes* (Ancylistaceae), *Completoria* (Completoriaceae), and *Orthomyces* (Entomophthoraceae) (Figure 2).

The quality check of the assembled data was carried out with MegaBLAST [32,33,34] by searching gene sequences against the non-redundant (nr) Genbank at NCBI (Bethesda MD:1 National Library of Medicine (U.S.); available from: https://blast.ncbi.nlm.nih.gov/Blast.cgi (accessed on 15 May 2013). Filtering and masking criteria were “Low complexity regions” and “Mask for lookup table only”. We only kept one strain per search incidence with hits generating an e-value of 0.0 and matches with 100% query coverage. Fungal sequences that did not fulfill these criteria were excluded from further analysis. For the analysis of LSU sequences, we used the D1/D2 domain of the 28S rDNA (which is 554 bp in length), a D-loop fragment of the 18S rDNA (894 bp), and an RPB2 fragment of about 777 bp in length.

### 2.2. Phylogenetic Analysis

We treated each of the ribosomal DNA sequence alignments (LSU and SSU) and the protein-coding RPB2 sequence alignment as separate data sets, which comprised 147, 91, and 71 sequences, respectively. The sequences were individually aligned for each locus using MAFFT version 7.123 [35], with the adequate algorithm being automatically chosen (FFT-NS-1, FFT-NS-2, FFT-NS-I, or L-INS-i; dependent on the data size). We selected BLOSUM 62 as the scoring matrix for amino acids. The parameters for the scoring matrix were 1 PAM/k = 2, as these parameters have been shown to be best suited for a data set focusing on closely related species [36]. We then visually inspected the multiple sequence alignments (MSAs) with BioEdit (Version 7.2.0, [37]), trimmed the beginning and end of each MSA, and manually excluded ambiguously aligned sections based on our visual inspection from further analyses based on 69 LSU, 54 SSU, and 36 rbp2 strains. Sequences of the genus *Neozygites* had to be excluded from the analyses as alignments with other entomophthoromycotinan sequences were not feasible due to uncharacteristically large sequence divergences. The resulting alignments are provided in Appendix A

The Bayesian inference and maximum likelihood approaches for phylogram reconstruction were calculated with the NSF (National Science Foundation, Arlington, VA, USA) supercomputer XSEDE [38] at CIPRES Science Gateway V. 3.1 (Cyberinfrastructure for Phylogenetic Research, www.phylo.org (accessed on 14 June 2013), [39]).

Statistical support was computed from 1,000 bootstrap replicates by using RAxML 7.0.4. [40,41] with the default models using GTRCAT for the bootstrapping phase and GTRGAMMA for the final tree inference. Bootstrap support was mapped automatically onto the best tree (majority rule extended) while we considered only support >90% as reliable.

Bayesian interference (BI) trees were estimated with MrBayes 3.1.2 [42] with default settings and the following changes: We ran five million generations for each analysis with a sample frequency of 0.25 and a sampling every 2000 trees. Posterior probability was collected with consensus majority rule and a “sumt burnin value” of 25.

The output files “infile.nex” (MrBayes) and “RAxML_bipartions” (RAxML) were visualized and re-rooted with Figtree version 1.4 (http://tree.bio.ed.ac.uk/software/figtree/) (accessed on 8 July 2013). Trees were edited and adjusted with Inkscape software (Version 0.91, Free Software Foundation, Inc. Boston, MA, USA).

### 2.3. Parasitic Fungi and Their Insect Hosts

In total, we investigated 429 strains encompassing 84 described entomophthoromycotinan species (Appendix A). Information on all known hosts of the Entomophthoromycota was obtained from the literature and the ARSEF database (10.15482/USDA.ADC/1326695). Insect host species records were summarized on an ordinal level. We only included data where the respective sampling site of a pathogenic strain could unambiguously be identified to a particular host. This allowed us to not only uniquely link affected host taxa with specific lineages of the Entomophthoromycota but also to estimate the number of parasitized species within these host taxa. Fungal species only collected in soil samples were not included here. A summarizing overview of all fungal-host associations is shown in Appendix A with information on the hosts taken from the ARSEF collection catalog (Agricultural Research Service, Collection of Entomopathogenic Fungal Cultures, Ithaca, NY, USA; Richard A. Humber) and from other data collections (NCBI; JMRC, FSU Jena, see above).

## 3. Results

### 3.1. Revised Phylogeny of the Entomophthoromycotina

Of the three analyzed gene sequences, we based the phylogeny of the Entomophthoromycotina on the LSU sequences, as they yielded the most comprehensive and robust phylogenetic backbone (compare Figure 3 with Appendix A for LSU, SSU, and RPB2, respectively).

The class Basidiobolomycetes is placed as a monophyletic sister group to the class Entomophthoromycetes at the basis of the Entomophthoromycotina (Figure 3). *Schizangiella* and *Basidiobolus*, the only known type genera in this class that were grouped in the family Basidiobolaceae, were also retrieved as monophyletic clades, respectively.

The second retrieved class is the Entomophthoromycetes, with the family Ancylistaceae clearly displayed as paraphyletic, containing five monophyletic branches, separated from the other two retrieved subfamilies in the Entomophthoromycetes, Entomophthoroidea, and Erynoidea. The first monophyletic ancylistacean branch almost exclusively contains species of the genus *Conidiobolus*, namely *C. coronatus, C. incongruus, C. firmipilleus, C. lamprauges, C. gonimodes, C. polytocus, C. brefeldianus*, *C. lichenicolus,* and *Microconidiobolus nodosus* (bootstrap support: 67%, Bayesian posterior probability: 1). We will refer to this lineage as Ancylistaceae *sensu stricto* in the following, which is in accordance with Gryganskyi et al. [1].

The second branch mainly contains species of *Batkoa* [1,2]. The obtained pattern implies the polyphyly of both *Batkoa* and *Conidiobolus* as traditionally defined (Figure 3) and strongly suggests that *Batkoa* is most closely related to *C. obscurus*.

Thus, merging clade D (*Batkoa* s.l. + Ancylistaceae II + Entomophthoraceae, Figure 3) into *Batkoa* appears to be justified. The third ancylistacean branch contains 30 species of *Neoconidiobolus*, *Capillidium,* and *Conidiobolus* but also *Entomophaga destruens*. This suggests that *Entomophaga* is not justified as a separate genus from a phylogenetic perspective. The monophyletic subfamily Entomophthoroidea is placed as a sister group to the clade Erynoidea. The former contains species of the genera *Entomophaga*, *Entomophthora*, *Massospora,* and *Eryniopsis*, whereas the latter contains the genera *Furia*, *Pandora*, *Erynia,* and *Zoophaga*.

The results do not confirm the monophyly of any of these genera; thus, their respective initial taxonomic classifications require further and more thorough revision. Besides the basal Basidiobolomycetes and the highly diversified Entomophthoromycetes, the postulated third class within Entomophthoromycotina, Neozygitomycetes, could not be phylogenetically assessed due to insufficient and inconclusive sequence data. It remains to be seen whether this taxon retains its status as a monophyletic class in future comparative studies. Also, two families within the class Entomophthoromycetes, Completoriaceae and Meristacraceae, could not be reliably included in the revised phylogeny because few type specimens and sequence data were available.

### 3.2. Patterns of Host Specificity of the Entomophthoromycotina

Of the two retrieved, monophyletic Entomophthoromycotina classes, Basidiobolomycetes are primarily soil dwellers and often observed at dung of amphibians, reptiles, and rodents, whereas Entomophthoromycetes reflect the highest species diversity with the broadest host spectrum for the parasitic taxa. This can partially be ascribed to the species assigned to the genus *Conidiobolus*, of which representatives are distributed among all retrieved Ancylistaceae clades. Particularly in insects, *Conidiobolus* host species range from various ametabolous and hemimetabolous orders (Collembola, Thysanoptera, Hemiptera, Isoptera) to different groups of the megadiverse Holometabola. Representatives of the exclusively entomopathogenic subfamily Entomophthoroidea, which contains the type species *Entomophthora muscae*, can be found as parasites in four distinctive and distantly related insect taxa, the hemimetabolous orders Orthoptera (only reported from the family Acridoidea so far) and Hemiptera (Cicadomorpha, Fulgoromorpha, Heteroptera, Sternorrhyncha), as well as in the two holometabolous orders Lepidoptera and Diptera (Figure 4). Species of the other retrieved entomopathogenic subfamily, Erynioideae, show a particularly strong tendency to infect species of the order Hemiptera but have also been reported to infect species from the orders Lepidoptera and Diptera (Figure 4, Appendix A). The class Neozygitomycetes is mono-ordinal (order Neozygitales), mono-familiar (family Neozygitaceae), and trigeneric, comprising the genera *Apterivorax*, *Neozygites* (Figure 4, Appendix A)*,* and *Thaxterosporium*, and appears to be specialized on mites and plant lice [1].

It was not possible to include the other previously characterized Entomophthoromycotina lineages (Figure 2) in both our phylogenetic revision and our host specificity assessment, either due to a paucity in exemplary specimens or insufficient sequence data. Thus, the phylogenetic relationships of Completoriaceae and Meristacraceae cannot be further resolved here. Concerning their respective host specificity, the only described species of Completoriaceae, *Completoria complens*, has been isolated as an intracellular parasite of fern gametophytes, whereas the two genera within the Meristacraceae, *Meristacrum,* and *Tabanomyces*, parasitize nematode worms and tabanid flies (Tabanidae, Diptera). 

### 3.3. New Type Species for Conidiobolus: C. coronatus

For the genus *Conidiobolus*, the lectotype *C. utriculosus* Brefeld 1884 [45] was indicated as a type for the genus by Clements and Shear in 1931 [46]. This species is lost and has been lectotypified by Nie et al. in 2020 [25] with some of Brefeld’s pictures, which were assigned to the species *C. minor*, which was also not cultivated. However, for the establishment of an unambiguous taxonomy, it is mandatory to access extant strains of the corresponding type species. Due to the prominence of *C. coronatus* as a human pathogen causing entomophthoromycosis [13], its global distribution and use as a model organism for fungal infections (as reviewed by Mendoza et al. [47]), we suggest maintaining the genus name *Conidiobolus* (which was recognized as *Conidiobolus* s.s. by Nie et al. [25]) and we propose *C. coronatus* as a new type for the genus *Conidiobolus* s.s.
Type: *Conidiobolus* Bref. 1884.Synonymy:*Boudierella* Costantin, *Bull. Soc. mycol. Fr.* 13: 40 (1897) [48].*Delacroixia* Sacc. and P. Syd., *Syll. fung.* (Abellini) 14(1): 457 (1899) [49].Classification: Entomophthoromycotina, Entomophthoromycetes, Entomophthorales, Ancylistaceae.
Comprises *Conidiobolus* clade C: Ancylistaceae I (see Figure 3).Note: Comparing *Conidiobolus coronatus* AF113418 with *Macrobiotophthora vermicola* AF052400 at the level of small subunit (18S, SSU) rDNA nucleotide sequences, the deviation was found to be high: identity = 824/967 (85%) and gaps = 34/967 (3%), which was lower than 90% SSU sequence similarity as proposed by [50] justifying classification in a new family.Lectotype: *Conidiobolus utriculosus* Bref. 1884 (Clements and Shear, Gen. Fungi: 239. 1931) [46].Proposed epitype: *Conidiobolus coronatus* (Costantin) A. Batko 1964 (typ. cons. prop.) [51] ≡ *Boudierella coronata* Costantin 1897 [48].Lectotype: Costantin, Bull. Soc. mycol. Fr. 13: 40 (1897): see Pl. 4, Figures 1–10, ; Pl. 5, Figures 11–17 in [48].Type: JMRC:SF:11506 (metabolically inactive culture), Sweden, isol. M. Gustafsson (B42536, here designated).Isotypes: NRRL 28638; CBS 209.66; ATCC 28846.Whole genome sequence PRJNA67455 available from: http://www.ncbi.nlm.nih.gov/bioproject/PRJNA67455 (accessed on 23 February 2016), see also [52].Reference sequences: AF113418 (SSU), AF113456, AY546691 and NG_027617 (LSU), AY997041 (ITS1-5.8S-ITS2), DQ275337 (elongation factor 1-alpha), DQ294591 (RNA polymerase II subunit RPB1), DQ302779 (RNA polymerase II subunit RPB2).

## 4. Discussion

The analyses of our sequence data confirm the monophyly of the subphylum Entomophthoromycotina with a broad taxon sampling, in agreement with Gryganskyi et al. [1] and Spatafora et al. [22]. Within the retrieved subphylum, however, discrepancies between the previously suggested phylogenetic relationships and our topology become apparent. For instance, the Ancylistaceae are retrieved as a tripartite paraphyletic clade (“Ancylistaceae I, II & III”) containing three monophyletic branches, mainly but not only composed of taxa currently assigned to *Conidiobolus*. Traditionally, this assemblage has been taxonomically unified based on several shared morphological features, such as a coenocytic mycelium, very small nuclei (2.5–4 μm), and a prominent central nucleolus. However, their monophyletic status has been questioned for decades, as well as their whole status as a taxonomic unit [1,53]. Our results now indicate that this group does indeed not constitute a monophyletic group. However, we strongly suggest that the well-known genus *Conidiobolus*, which also contains human-pathogenic strains, should be conserved. Based on our phylogenetic assessments and previous data, we propose *Conidiobolus coronatus* as the new type for the genus *Conidiobolus* sensu stricto.

Concerning host specificity, we thoroughly investigated all relevant species and strains kept in fungal collections and performed an exhaustive literature survey. The host spectrum of all investigated taxa varies tremendously within the Entomophthoromycota, reflecting a plethora of different lifestyles unified in this clade (Figure 4). Far from being exclusively restricted to insects, hosts of the Entomophthoromycota range from algae (*Ancylistes*) over nematodes (*Macrobiotophthora*) and mites (*Neozygites*), through different types of vertebrates up to mammals (*Basidiobolus* and *Conidiobolus*) (Figure 4). Clearly, the complete host spectrum of this subphylum is far from being exhaustively resolved, as the infection capacity and host specificity of many taxa remain unknown. This is underlined by our survey on all known host species from the thus far described taxa within the Entomophthoromycota. For over 50% of the investigated fungal species, it is not known whether they have a parasitic life, and if so, which specific hosts they can infect (Figure 4).

Focusing solely on the eponymous entomopathogenic species, it becomes apparent that host specificity and host spectra are very unevenly distributed among the investigated taxa, potentially also hinting at sampling biases. In addition, there is no recognizable congruence in the phylogeny of insects and the two primarily entomopathogenic subfamilies of the Entomophthoromycotina, i.e., Entomophthoroidea and Erynoidea, with parasitism apparently having evolved many times independently within them. Despite the limitations of our knowledge of the real extent of their respective host spectra, the asymmetries of the thus far described host specializations are striking. For instance, the highly diverse insect order Hemiptera (including plant lice, cicada, true bugs, and moss bugs) contains by far the most host species for all investigated classes of the Entomophthoromycotina except the primarily saprobic Basidiobolomycetes. Diptera, Lepidoptera, and Orthoptera are also among the most frequently parasitized insect orders, with the latter exclusively serving as hosts for species of the subfamily Entomophthoroideae according to the current state of knowledge.

Entomophthoromycotina belongs to the oldest known terrestrial lineages of fungi, with an estimated appearance in the Silurian more than 400 mya [1,20,21]. The earliest Pterogyta (i.e., winged insects) fossils (Palaeodictyoptera) are known from the early Carboniferous, 358 mya [54]. As the Pterogyta constitute the most dominantly parasitized insect host group of the Entomophthoromycotina (Figure 4), this hints at a potential correlation of the diversifications in both groups. However, other factors potentially accounting for asymmetric host preference patterns have to be taken into consideration as well, such as optimized dispersal and different defense mechanisms of host species. In any case, the dramatic rise of pterygote insects has very likely increased the dispersal capacities for associated entomopathogenic fungi, far beyond the hypothesized ancestral way of dispersal via apterygote insects primarily living in soil and leaf litter. Generalized saprotrophic soil dwellers are still often linked to hosts such as Collembola (springtails) [55], or other arthropods living in the ground, e.g., mites (Acari). Even though it becomes apparent from our study that there is no strict link between insect lifestyles and specific fungal families, we hypothesize that the appearance and eventual dominance of winged insects largely broadened the spectrum of available hosts for the entomopathogenic lineages of the Entomophthoromycotina. Furthermore, it has been suggested that the ability to digest chitin, which constitutes an efficient infection barrier, had most likely already evolved prior to the radiation of winged insects in this very old lineage of terrestrial fungi [10,21,53]. Concordantly, similar infection patterns within Entomophthoromycotina hint at their ancestral potential to infect insect hosts, and this repeatedly occurred and in similar ways during the evolution of different lineages within this subphylum [56,57]. That no recognizable congruence between the phylogenic patterns in insects and Entomophthoromycotina could be discerned strongly suggests that parasitism evolved many times independently within this clade from saprobic ancestors (Figure 4).

In conclusion, we confirm the overall monophyly of the phylogenetically controversial fungal clade Entomophthoromycotina. Whereas certain subgroups were supported as monophyletic in our analyses, others, especially in the Ancylistaceae lineages, have to be critically re-evaluated. The previously established phylogenetic concept of the prominent taxon *Conidiobolus* cannot be upheld, and we propose *C. coronatus* as a new type for this genus. Our extensive survey of host specificity within parasitic lineages of the Entomophthoromycotina does not show clear patterns of co-evolution with their primary targets, insects, but suggests that the radiation of winged insects specifically favored their dispersal. This opens up the avenue for future studies expanding the investigations on the complete host spectrum of this subphylum, potentially also contributing to their successful application as biological pest control agents.

## Figures and Tables

**Figure 1 microorganisms-10-00256-f001:**
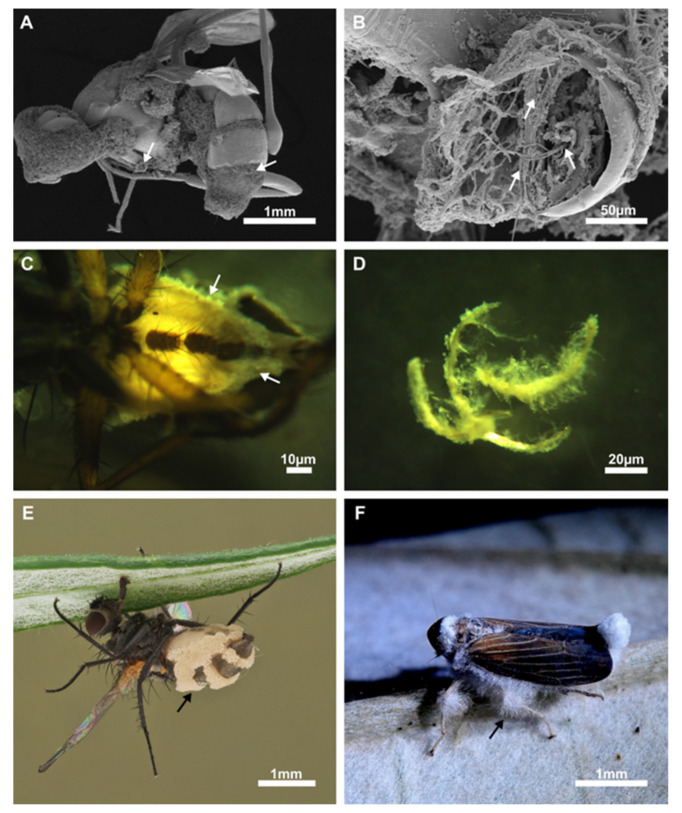
Examples of host infections by the Entomophthoromycotina. (**A**) Scanning electron microscopic image of *Meroplius fasciculatus* (Sepsidae, Diptera) with a substantial fungal infection (*Entomophthora* sp.). Fungal outgrowth points are indicated by arrows. (**B**) Separated prothoracic leg of *Meroplius fasciculatus* (Sepsidae, Diptera) shows the extent of fungal growth within the host tissue (arrows). (**C**) Light microscopic image of the abdomen of *Meroplius* sp. (Sepsidae, Diptera) with an infection of *Entomophthora muscae* (Entomophthoraceae) along the pleurae, less sclerotized structures that potentially facilitate outward growth of the fungus (arrow). (**D**) Isolated hyphae of the fungus in (**C**) strongly hint at an intimate somatic contact to the pleura of the host insect. (**E**) Macro camera image of an infected *Delia* sp. (Anthomyiidae, Diptera) with an entomophthoralean infection, arrows indicate fungal hyphae and conidia protruding from the abdomen of the host insect. (**F**) Macro camera image of an Ecuadorian leafhopper from the family Cicadellidae (Cicadomorpha, Auchenorrhyncha) with entomophthoralean hyphae growing from multiple segments of its infected body. Images (**A**–**D**,**F**): by Lars Möckel, Image (**E**): by Hans Pohl (Friedrich Schiller University Jena, Germany).

**Figure 2 microorganisms-10-00256-f002:**
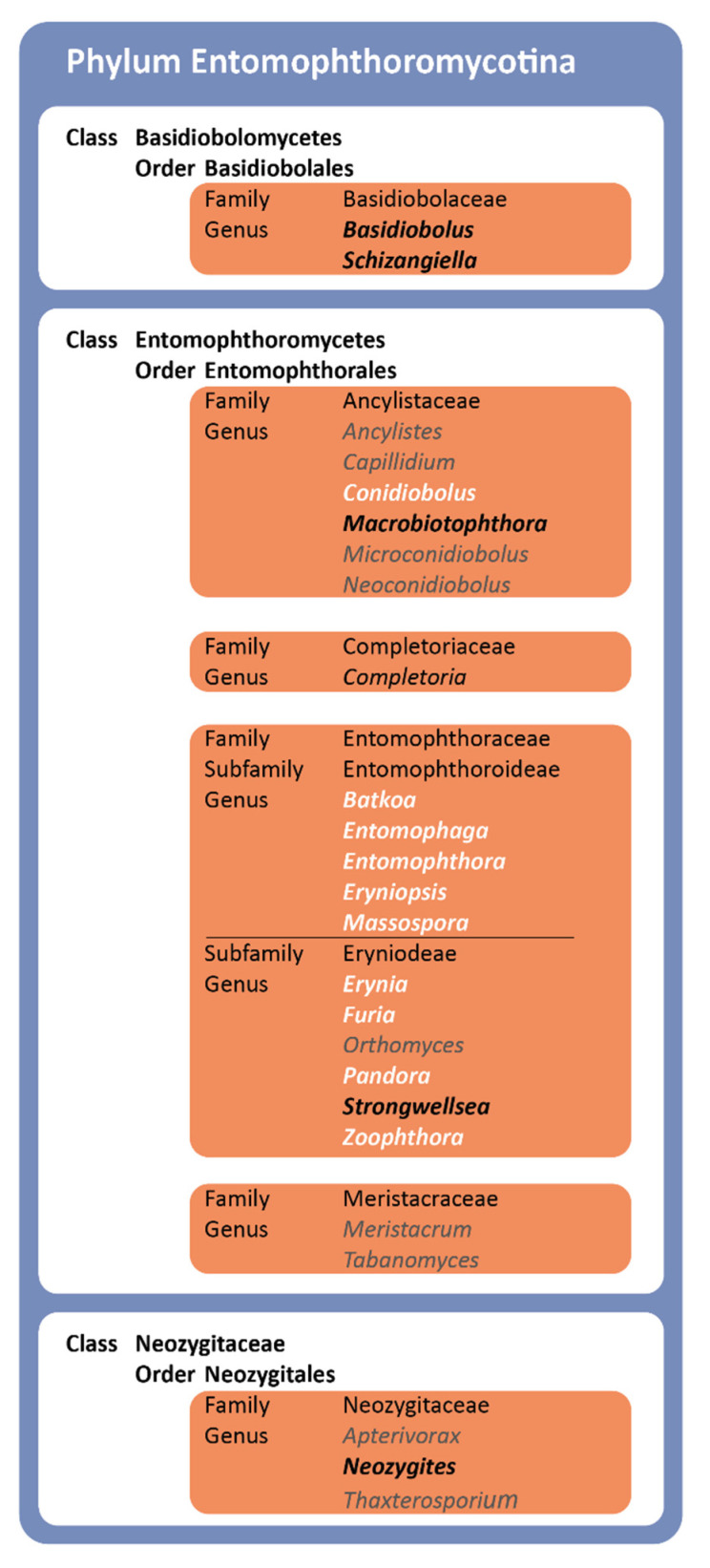
Entomophthoromycotina taxonomy based on the system of Humber [31]. Names in black indicate taxa represented by strains in the present study with available rRNA sequences from the International Nucleotide Sequence Database Collaboration (INSDC). For taxa in gray, sequences were not available. Concerning genera, names in bold white indicate the availability of both LSU and SSU sequences for their representative strains, whereas names in bold black indicate genera for which only SSU sequences were available.

**Figure 3 microorganisms-10-00256-f003:**
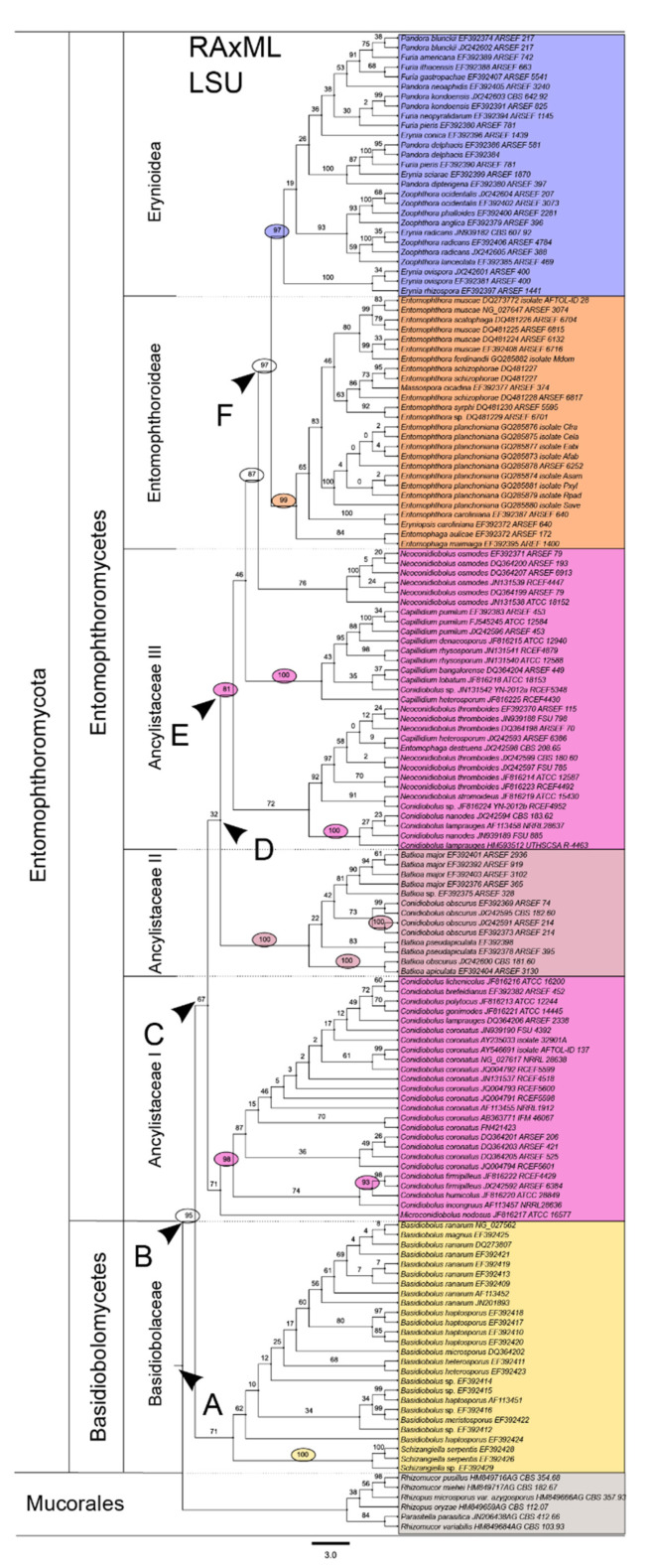
Entomophthoromycotina phylogeny based on a maximum likelihood phylogram reconstruction inferred from large subunit (28S, LSU) ribosomal RNA nucleotide sequences of 147 taxa. Statistical bootstrap support (BS, inferred with RAxML) is indicated above branches. BS above 90%, above 80% and 70% were considered as high, well, and moderate support in accordance to earlier published rRNA phylogenies of entomophthoralean and mucoralean species, respectively [43]. Well-supported branches are marked with ellipses in the color code of the corresponding clade. Six species of Mucorales were collectively used as an outgroup. (**A**) Split of Mucorales (outgroup) + Entomophthoromycotina, (**B**) Split of Basidiobolomycetes + Entomophthoromycetes, (**C**) Split of Ancylistaceae I + remaining clades of Entomophthoromycetes, (**D**) Split of *Batkoa* s.l. + Ancylistaceae II + Entomophthoraceae, (**E**) Split of Ancylistaceae II + Entomophthoraceae, (**F**) Split of Entomophthoroideae + Erynoideae. Scale bar indicates substitutions per site.

**Figure 4 microorganisms-10-00256-f004:**
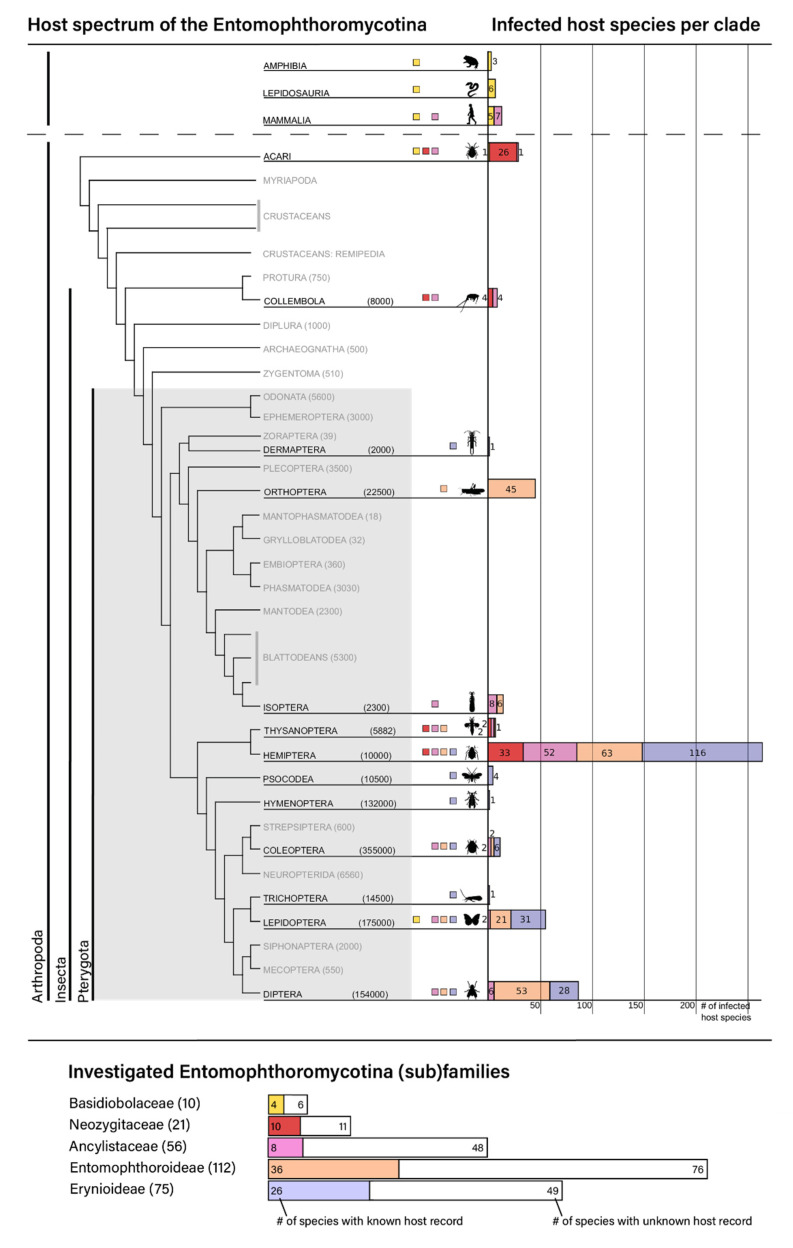
Representation of the host spectrum of the Entomophthoromycotina with particular emphasis across the phylogeny of the Insecta. The occurrence of entomophthoromycotan (sub)families is mapped next to their respective host taxa. Different colors represent the different (sub)families of the Entomophthoromycotina with documented host records. Numbers in brackets indicate the documented insect species per clade, whereas, on the right-hand side, the numbers of host species known to be infected by the entomophthoromycotan (sub)families per clade are indicated. The primarily infected Pterogyta are shaded in light grey. The arthropod phylogeny has been adapted from Misof et al. [3] and Beutel et al. [44] and was further modified by collapsing the nodes to the ordinal levels. Representative pictograms indicate characteristic insect host taxa; insect orders without records of host species are in gray capitals. Classification of fungal families after Humber [31]. The entomophthoromycotan taxa are indicated on the bottom with their respective total numbers of described species (right-hand side, 274 to date) vs. the number of species with a documented host record left-hand side, 84 to date). Detailed information on all fungal-host interactions are given in Appendix A.

## Data Availability

All data underlying the presented study will be made available at an online data repository.

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
