# Peer review of "Phylogenetic Revision and Patterns of Host Specificity in the Fungal Subphylum Entomophthoromycotina"

_microorganisms, 2022, doi:10.3390/microorganisms10020256_

Round 1

Reviewer 1 Report

Dear Authors,

Thank you for the opportunity to review your manuscript. It is solid work and a well-done analysis.  I have a few small considerations. 

Several minor mistakes through the text and misspelled species names were detected, page 8, lines 247, 256, 291 as examples.  I would recommend thorough spellcheck through the text before publishing. Also, A would recommend using current names for the genera and species of Capillidium, Neo- and Macroconidiobolus.

Respectfully,

Reviewer.

Author Response

Dear Authors,

Thank you for the opportunity to review your manuscript. It is solid work and a well-done analysis.  I have a few small considerations. 

Several minor mistakes through the text and misspelled species names were detected, page 8, lines 247, 256, 291 as examples.  I would recommend thorough spellcheck through the text before publishing.

Thank you for pointing us to these. We performed a thorough spell-check and are now confident to have located and corrected all misspelled terms.

Also, A would recommend using current names for the genera and species of Capillidium, Neo- and Macroconidiobolus.

We now have adjusted all genus names as suggested in the main text as well as in the figures (2 and 3) to bring them up to date.

Respectfully,

Lars Möckel

Reviewer 2 Report

I am glad to see that the present work revealed a new and valuable phylogenetic insight of entomophthoroid fungi. This manuscript reconstructed a phylogenetic analysis including 159 entomophthoroid fungi based on sequence data of three loci, and confirm the monophyly of this fungal group once more. We noted that Entomophaga destruens (CBS 208.65) was located in the “Ancylistaceae III”, but would you think about that this strain had a wrong identification, the same situaiton was also found in Conidiobolus lamprauges (ARSEF 2338) and Conidiobolus heterosporus (ARSEF 6386). Meanwhile, Conidiobolus pseudapiculatus was reassigned to the genus Batkoa, and Conidiobolus antarcticus is synonymised with Conidiobolus osmodes.

Particularly, a new type of Conidiobolus coronatus is recongized for the genus Conidiobolus s.s. to replace C. utriculosus, this is a very valuable point. However, this taxonomic treatment should be voted after the Nomenclature Committee for Fungi, and the type specimen here is not collected and isolated from the original location of C. utriculosus. Furthermore, I suggest more available ex-types with molecular data should be used in the phylogenetic analyses. Such as more species in Capillidium and related genera. The genus Batkoa maybe recognized as a new family in the future rather than place it into “Ancylistaceae II”. On the other hand, I agree your point that no apparent co-evolutionary events with any particular host lineage associated with related entomophthoroid fungi, and winged insects may play a positive effect on the widespread of entomophthoroid fungi.

  1. Result of 3.3

Our recent taxonomic revision of Conidiobolus should be stated detailed.

  1. Line 358

“which was recognized as Delacroixia by Nie et al. ” should be rephrased to “which was recognized as Conidiobolus s.s. by Nie et al. ”

  1. Line 359

Conidiobolus” changed to “Conidiobolus s.s.”

  1. Line 373

“Proposed new type” changed to“epitype”

  1. Line 388 and 389

 Please delete “Gryganskyi et al.”

Author Response

I am glad to see that the present work revealed a new and valuable phylogenetic insight of entomophthoroid fungi. This manuscript reconstructed a phylogenetic analysis including 159 entomophthoroid fungi based on sequence data of three loci, and confirm the monophyly of this fungal group once more. We noted that Entomophaga destruens (CBS 208.65) was located in the “Ancylistaceae III”, but would you think about that this strain had a wrong identification, the same situaiton was also found in Conidiobolus lamprauges (ARSEF 2338) and Conidiobolus heterosporus (ARSEF 6386). Meanwhile, Conidiobolus pseudapiculatus was reassigned to the genus Batkoa, and Conidiobolus antarcticus is synonymised with Conidiobolus osmodes.

Thank you very much for this advice.

Particularly, a new type of Conidiobolus coronatus is recongized for the genus Conidiobolus s.s. to replace C. utriculosus, this is a very valuable point. However, this taxonomic treatment should be voted after the Nomenclature Committee for Fungi, and the type specimen here is not collected and isolated from the original location of C. utriculosus. Furthermore, I suggest more available ex-types with molecular data should be used in the phylogenetic analyses. Such as more species in Capillidium and related genera. The genus Batkoa maybe recognized as a new family in the future rather than place it into “Ancylistaceae II”. On the other hand, I agree your point that no apparent co-evolutionary events with any particular host lineage associated with related entomophthoroid fungi, and winged insects may play a positive effect on the widespread of entomophthoroid fungi.

  1. Result of 3.3

Our recent taxonomic revision of Conidiobolus should be stated detailed.

We thank for this valuable advice regarding the recognition of Conidiobolus coronatus as type for the genus Conidiobolus s.s. from a nomenclatural point of view. We do completely agree with the concerns raised by the reviewer with respect to the vote by the Nomenclature Committee for Fungi (NCF). A corresponding conservational proposal which rejects C. utriculosus and proposes C. coronatus as new type for Conidiobolus s.s. which will include an official review by the NCF is currently in preparation for the journal Taxon.

With respect to the geographic origin of the newly proposed type, we are aware of the difficulties of deviations in the geographic origins of the original lectotype with the newly proposed type. The lectotype Conidiobolus utriculosus Bref. 1884 (Designated by Clements & Shear, Gen. fung., Edn 2 (Minneapolis): 239, 1931) was originally isolated from Germany. The newly proposed type JMRC:SF:11506 and its isotypes NRRL 28638, CBS 209.66, ATCC 28846 originate from Sweden.

There are five more candidates for putative ex type strains which are morphologically well-characterized and freely available from commercial culture collections worldwide. These are:

CBS 140 26 unknown geographic origin

CBS 176.55 Italy

CBS 647.68 Netherlands

CBS 110.76 Netherlands

CBS 183.90 France

All these strains originate in Europe but not in Germany. A search at the American Type Culture Collection (ATCC) revealed similar results. None of the C. coronatus strains originate from Germany.

In order to designate a type which is morphologically and genome biologically well-characterized and easily accessible (deposited in a wide variety of collections worldwide) we decided to vote for the type as described and proposed here.

The genome of JMRC:SF:11506 and its isotypes NRRL 28638, CBS 209.66, ATCC 28846 was elucidated and extensively studied by Chang et al. (Genome Biol. Evol. 2015, 7, 1590-1601, doi:10.1093/gbe/evv090). This guarantees reproducibility and provides a firm basis for future analyses in the scientific community.

Due to lack of access to ex type strains at the beginning of this study we were not able to include any more, and decided to perform a profound revision of the taxonomic names and their synonyms in accordance to the paper from Nie et al. 2020 instead (MycoKeys 2020, 66, doi:10.3897/mycokeys.66.46575).

  1. Line 358

“which was recognized as Delacroixia by Nie et al. ” should be rephrased to “which was recognized as Conidiobolus s.s. by Nie et al. ”

Thank you for recognizing this mistake from our side, we have changed the text accordingly.

  1. Line 359

Conidiobolus” changed to “Conidiobolus s.s.”

Thank you, we changed this accordingly.

  1. Line 373

“Proposed new type” changed to“epitype”

Corrected

  1. Line 388 and 389

 Please delete “Gryganskyi et al.”

Thank you, deleted and changed accordingly.

Reviewer 3 Report

Entomophthoroid fungi, an important group of early diverging fungi or basal fungi or lower fungi, are saprophytic and also associated with insects. Entomophthoromycota was erected by Humber in 2012 to encompass this monophyletic group of fungi. Implied by genomic information, it is treated as subphylum Entomophthoromycotina by Spatafora et al. in 2016. This article further confirms its monophyly, based on several molecular markers from numerous strain materials. Meanwhile, this article provides insights into the relationship within this subphylum and clarifies multiple and independent origins of parasitism patterns of co-evolution across their hosts. This work will enhance our understanding of the phylogeny and evolution of this important group of basal fungi. As far as the paraphyletic grade Ancylistaceae is concerned, it is grouped into five distinct clades. Therefore, it is suggested to discuss this family in five parts rather than three, or to propose some new families. Some text improvements are suggested as follows.

  1. Lines 184-185

“The Bayesian inference, Maximum Likelihood approaches and phylogenetic tree Inference” is suggested as “The Bayesian inference, Maximum likelihood approaches for phylogram reconstruction”.

  1. Line 243

“Entomophtho-romycotina” should be “Entomophthoromycotina”

  1. Line 358

The “Conidiobolus (which was recognized as Delacroixia by Nie et al. [25]” needs to be revised as “Conidiobolus following the treatment Nie et al. [25]”

  1. Legends of figures S1-S6

All the first words “Phylogeny” should be replaced by “Phylogeny of”.

Author Response

Entomophthoroid fungi, an important group of early diverging fungi or basal fungi or lower fungi, are saprophytic and also associated with insects. Entomophthoromycota was erected by Humber in 2012 to encompass this monophyletic group of fungi. Implied by genomic information, it is treated as subphylum Entomophthoromycotina by Spatafora et al. in 2016. This article further confirms its monophyly, based on several molecular markers from numerous strain materials. Meanwhile, this article provides insights into the relationship within this subphylum and clarifies multiple and independent origins of parasitism patterns of co-evolution across their hosts. This work will enhance our understanding of the phylogeny and evolution of this important group of basal fungi. As far as the paraphyletic grade Ancylistaceae is concerned, it is grouped into five distinct clades. Therefore, it is suggested to discuss this family in five parts rather than three, or to propose some new families. Some text improvements are suggested as follows.

  1. Lines 184-185

“The Bayesian inference, Maximum Likelihood approaches and phylogenetic tree Inference” is suggested as “The Bayesian inference, Maximum likelihood approaches for phylogram reconstruction”.

Thank you, we corrected this passage accordingly.

Line 243

“Entomophtho-romycotina” should be “Entomophthoromycotina”

Thank you, we corrected this spelling error.

Line 358

The “Conidiobolus (which was recognized as Delacroixia by Nie et al. [25]” needs to be revised as “Conidiobolus following the treatment Nie et al. [25]”

Thank you for recognizing this misapplication of the reference, which we have now changed accordingly.

  1. Legends of figures S1-S6

All the first words “Phylogeny” should be replaced by “Phylogeny of”.

Corrected